# Effect of Multi-Element Microalloying on the Structure and Properties of High Chromium Cast Iron

**DOI:** 10.3390/ma16093292

**Published:** 2023-04-22

**Authors:** Tao Liu, Jibing Sun, Zhixia Xiao, Jun He, Weidong Shi, Chunxiang Cui

**Affiliations:** 1Key Laboratory for New Type of Functional Materials in Hebei Province, School of Materials Science and Engineering, Hebei University of Technology, No. 5340 Xiping Road 1, Beichen District, Tianjin 300401, China; 18406580738@163.com (T.L.); xiaozhixia@hebut.edu.cn (Z.X.); 13512452879@163.com (J.H.);; 2Tianjin Lixinsheng New Material Technology Co., Ltd., Tianjin 301602, China

**Keywords:** iron alloys, high chrome cast iron, alloying, mechanical property

## Abstract

High chromium cast iron (HCCI) has been widely used as wear-resistant material in the industry. Alloying is an effective way to improve the microstructure and mechanical properties of HCCI. This paper added multi-component V-Fe-Ti-Nb-C-Zr-B alloy (VFC) to HCCI, showing a significant synergistic solution-strengthening effect. The results show that the added V-Ti-Nb-B are dissolved in M_7_C_3_ carbide to form the (Cr, Fe, V, Ti, Nb)_7_(C, B)_3_ alloy carbide, and a small amount of V and all Zr are dissolved in austenite and martensite. Adding VFC into HCCI improved the hardenability of HCCI, decreased the residual austenite content from 6.0 wt% to 0.9 wt%, increased the martensite content from 70.7 wt% to 82.5 wt%, and changed the structure and content of M_7_C_3_ carbide. These changes increased the hardness of as-cast and heat-tread HCCI by 1.4% and 4.1%, increased the hardness of austenite and martensite by 7.9% and 7.0%, increased the impact toughness by 16.9%, and decreased the friction coefficient and wear loss by 2.3 % and 7.0 %, respectively. Thus, the hardness, toughness, wear resistance, and friction resistance of HCCI alloy are improved simultaneously.

## 1. Introduction

High chromium cast iron (HCCI) has much higher wear resistance and is widely used in manufacturing wear-resistant parts [1,2]. HCCI consists of the essential components of Fe, Cr, and C, and its microstructure is composed of martensite or austenite and M_7_C_3_ carbide. M_7_C_3_ carbide possesses high hardness (1300–1800 HV), and its chemical composition, quantity, and morphology are mainly determined by the alloying elements in HCCI [3]. Researchers [4,5,6,7,8] have studied the role of alloying elements in HCCI. After adding 1.19 wt% V to HCCI containing 19 wt% Cr, M_7_C_3_ content increased from 30.97 vol% to 31.96 vol%, the carbide size decreased from 7.48 μm to 6.74 μm, and M_6_C_5_ carbide is formed [4]. The MC carbide formed when V content was higher than 2 wt% [5]. VC has a higher hardness (2600–3000 HV) and certain plasticity than the 1300–1800 HV of M_7_C_3_ carbide in HCCI. Kopycinski et al. [9] added 0–0.2 wt% Ti to HCCI (Cr 21.2 wt%, C 1.7 wt%). The bending strength of as-cast HCCI increased from 821MPa to 995MPa with the addition of 0.1 wt% Ti and decreased to 987 MPa with the addition of 0.2 wt% Ti. When 0.97 wt% Ti was added to HCCI containing 28 wt% Cr and 3 wt% C, the primary carbide size decreased from 1 mm to 50–350 μm [6,7]. The formed TiC can be used as the heterogeneous nucleation center of Cr_7_C_3_, refining M_7_C_3_ carbide significantly. NbC formed when Nb was added to hyper-eutectic HCCI containing 19–20 wt% Cr and 2.8 wt% C. NbC distributed at the grain boundaries can hinder the growth of primary M_7_C_3_ carbide and refine the M_7_C_3_ carbide size. Maja et al. [10] added 0.1–0.4 wt% Nb to HCCI (Cr 10–30 wt%, C 2–3.5 wt%) and observed that the hardness of as-cast HCCI increased from 432 HB to 485 HB. After adding 0.4–0.6 wt% Nb and heat treatment, the hardness of cast iron slightly decreases, but the bending strength increases by 40% [8]. With increasing the Nb content from 0 to 1.79 wt%, the formed NbC perform a second phase strengthening role in HCCI containing 15 wt% Cr and 3 wt% C and increases the hardness of as-cast HCCI from 48 HRC to 55 HRC [11]. Zr is a strong carbide-forming element in steel. Additionally, Zr has a high binding affinity with C, O, and N and can perform a role in the deoxidation and purification of molten steel [12]. According to the Fe-Zr phase diagram [13], the solubility of Zr in α-Fe and γ-Fe is 0.1 at% and 0.2 at%, respectively. When Zr forms the ZrO_2_ in steel [14] or precipitates as (Fe, Zr) at the ferrite/martensite boundaries [15], these precipitates can hinder the growth of austenite grains. Adding 0.11 wt% Zr to ferrite/martensitic two-phase steel containing 19 wt% Cr and 0.2 wt% C [15] can reduce the austenite grain size from 15.8 μm to 12.8 μm. B can be solidly dissolved in the M_7_C_3_ carbide of HCCI to increase the hardness of M_7_(C, B)_3_ [16]. Adding 1.5 wt% B can improve the wear resistance of HCCI while adding 3 wt% B harms the wear resistance of HCCI [17].

Compared with adding one alloying element to HCCI, adding multiple alloying elements to HCCI can better refine the microstructure and improve the mechanical properties [5,18]. Bedolla et al. [5] added 2.02 V-1.84 Ti-1.95 Nb (wt%) to HCCI, decreasing the wear loss by 21.5%. After adding V-Ti-Nb-Mo alloy to HCCI, You et al. [18] found that V can dissolve in M_7_C_3_ carbide to improve its hardness by solution strengthening and can form VCr_2_C_2_ compound, TiC and NbC particles were precipitated in austenite, and Mo was soluted in austenite to improve the hardenability of the alloy. Finally, adding 0.60V-0.60Ti-0.60Nb-0.35Mo (wt%) alloying elements increased the toughness and hardness of cast iron by 45.4% and 13.8%, respectively, and the synergistic effect of these elements led to the fine eutectic structure.

According to the heat treatment processes in references [19,20,21,22], for HCCI containing 2.76% C and 17.91% Cr, the commonly used quenching process is 960 °C × 3 h, tempering process is 450 °C × 2h.

Therefore, adding a strong carbide-forming element can improve the properties of HCCI alloys through a solution or second-phase strengthening. Multi-element microalloying has better synergistic strengthening effects. However, few people have studied the following two scientific issues so far. One is that Zr has a more vital ability to form carbides than Ti, Nb, V, Cr, and Fe, which is rarely considered when adding multiple alloying elements. The second is that the added multiple alloying elements mainly form different carbides. Few people have discovered and discussed the impact of these alloying elements on the crystal structure of M_7_C_3_. V is the most effective alloying element. According to reference [5], we added a high content of V and a small amount of C to the multi-element alloy, hoping to compare the effect of MC and M_2_C phases formed by adding V alone in the ingot. Simultaneously, according to the specific content of microalloying elements added in references [4,5,6,7,8,9,10,13,16,18], multi-element 75.9V-12.6Fe-5Ti-3.1Nb-2.2Zr-0.6C-0.6B alloy is added to HCCI to explore multiple synergistic strengthening effects.

## 2. Materials and Methods

### 2.1. Alloy Preparation

A 75.9V-12.6Fe-5Ti-3.1Nb-2.2Zr-0.6C-0.6B (for short, VFC) ingot was vacuum arc-melted using pure Fe, V, Ti, Nb, Zr, pig iron, and Fe-17 wt% B alloy. The HCCI ingot with a composition of 2.76C-0.27Si-0.52Mn-17.91Cr-0.12Mo-0.22Ni (wt%) was prepared using the medium-frequency induction melting furnace. New alloyed HCCI (for short, HCCI-VFC) was prepared by melting the HCCI ingot with 3.1 wt% VFC ingot (=2.29V-0.46Fe-0.15Ti-0.10Nb-0.07Zr-0.02C-0.02B) (wt%) in the medium-frequency induction melting furnace. HCCI and HCCI-VFC alloys are vacuum-heated at 960 °C for 3 h, cooled to room temperature, tempered at 450 °C for 2 h, then cooled to room temperature in the furnace.

### 2.2. Structural and Mechanical Property Test

Rigaku Dmax 2500Pc X-ray diffractometer was used for phase characterization, and the XRD results were refined using FullProf software (version: January-2023) to calculate the cell parameters and phase content. Then, a Hitachi S-4800 scanning electron microscope (SEM) was used to observe the microstructure, the KathMatic KC series laser confocal microscope and SEM were used to analyze the fracture morphology, and the Nano Measurer software (version: 1.2.5) was used to calculate the grain size. The samples were etched with 5 % nitrate alcohol for 40 s before SEM observation.

We used the HR-150A Rockwell hardness tester to test the hardness of samples, take the average value of five tests on each sample, and used the HWV-2T Vickers hardness tester to test the microhardness. According to GB/T229-2007 of Chinese national standard, the impact sample size was processed to 10 mm × 10 mm × 55 mm. We used the JB-200 pendulum impact testing machine to test the impact toughness at 150 J impact energy and take the average value of three samples. According to GB/T3960-2016 standard, the wear samples were processed to “30 mm × 7 mm × 6 mm”—in engineering drawings of all sizes, and the M-200 model friction and wear testing machine was used to conduct the wear resistance test. The counterpart’s materials in the sliding friction and wear test are GCr15 bearing steel. The loading test force was 350 N, the rotation speed of the friction pair was 200 r/min, and the test time was 1 h. The weight loss was measured using an analytical balance with an accuracy of 0.00001 g, and the wear loss averaged the three tests.

## 3. Results and Discussion

### 3.1. Microstructure

Figure 1a–d shows OM metallographic photos of as-cast and heat-treated HCCI and HCCI-VFC. The microstructure of as-cast HCCI is composed of elliptical austenite and M_7_C_3_ carbide distributed at grain boundaries. After heat treatment, fine particles or fine lath eutectic carbide at the grain boundaries and strip- or block-like primary carbide form a network structure. Figure 1e is an SEM image of the as-cast HCCI alloy composed of M_7_C_3_ carbide, austenite (for short, A), and a small amount of martensite (for short, M). According to the EDS results in Table 1, region A has a composition of (Cr_0.51_Fe_0.46_M_0.03_)_7_C_3_ (M_0.03_ = Si_0.004_Mn_0.022_Ni_0.004_) without V, Ti, and Nb, and corresponds to the eutectic M_7_C_3_ carbide with a petal microstructure composed of granular (~1.0 μm) and rod-shaped (4.6 ± 1.8 μm long and 1.1 ± 0.2 μm wide) grains. Region B is a blocky primary M_7_C_3_ carbide with a composition of (Cr_0.49_Fe_0.45_M_0.07_)_7_C_3_ (M_0.07_ = Si_0.021_Mn_0.021_Ni_0.013_Mo_0.004_), rich in Si and Mo. Its average size is 6.1 ± 2.6 μm, and some are more than 10 μm long. Region C presents a nearly hexagonal shape and has a composition of (Fe_0.83_Cr_0.13_M_0.04_)C_x_ (M = Si, Mn, Mo, Ni) and a Vickers hardness of 430.7 HV of austenite in HCCI (400–500 HV). The average grain size of austenite in HCCI ingot is 18.6 ± 13.3 μm, including large grains of 28.0 ± 9.8 μm, as shown in a red circle, and small grains of 7.5 ± 2.5 μm, as shown in purple circles.

Figure 1f–h shows the SEM images of HCCI after heat treatment. Figure 1f shows that as-quenched HCCI is composed of M_7_C_3_ carbide, M, and a small amount of residual austenite (for short, A’). Region D is M_7_C_3_ carbide with a composition close to (Cr_0.52_Fe_0.44_M_0.04_)_7_C_3_ (M_0.04_ = Si_0.022_Mn_0.005_Ni_0.013_). Region E is lath M grain with a composition close to (Fe_0.81_Cr_0.14_M_0.05_)C_x_ (M = Si, Mn, Mo, and Ni). Vickers hardness of region E is 690.3 HV, proving that the matrix has changed from A to M. Figure 1g shows that the fine eutectic carbide and the long-strip or blocky primary carbide are distributed on the grain boundaries, forming a network structure in the tempered HCCI. The granular eutectic carbide is ~1 μm, and the stipe-shaped eutectic carbide is 5.5 ± 2.7 μm long and 1.0 ± 0.2 μm wide. EDS results show that the matrix composition in region F of tempered HCCI alloy is similar to that in region C but slightly rich in Mn. The averaged Vickers hardness of region F is 713.4 HV, improved by 23.1 HV, meaning that a part of A’ has transformed into M after tempering, and the increased M content further increases the hardness of the HCCI matrix. In addition, the average grain size of the tempered alloy is 18.1 ± 8.5 μm, equivalent to as-cast alloy. Additionally, Figure 1h shows that white granular secondary carbides are dispersed on the matrix. Region G is rich in Fe, Cr, B, and C (Table 1). Theoretically, it is possible to form M_3_C, M_7_C_3_, and M_23_C_6_ carbide. According to EDS results, the white granular secondary carbide should be M_7_C_3_ carbide.

Figure 1i shows the SEM image of the VFC alloy. The ingot shows three contrasts. Black region H is rich in V, Ti, and C, and its composition is (V_68.8_Zr_0.1_Ti_3.9_Fe_0.7_Nb_1.5_)C_25.0_ = (V, M)_3_C. V-C compounds have no stable V_3_C structure, while V_2_C, Fe_2_C, and Nb_2_C have similar stable M_2_C structures. Therefore, the added trace elements can dissolve in the V_2_C structure to form a (V, M)_2_C alloy solid solution. Gray region I is rich in V and Fe, corresponding to the V (Fe, M) matrix. White region J is (V_3.4_Zr_29.9_Ti_11.1_Fe_0.6_Nb_1.5_)(B_18.2_C_35.3_) = (Zr, M)_0.9_(B, C), rich in Zr, B, and C. ZrC, VC, TiC, and NbC can form MC structures. Therefore, V, Ti, and Nb replace Zr, and B replaces C to form the (Zr, M)(B, C) phase.

Figure 1j is the SEM image of the as-cast HCCI-VFC alloy. It is also composed of M_7_C_3_ carbide, A, and a small amount of M. Region K has the composition of (Cr_0.54_Fe_0.43_M_0.03_)_7_C_3_ (M_0.03_ = Si_0.009_Mn_0.015_Ni_0.006_), containing 0.5 at% V, 0.1 at% Ti, and 0.1 at% Nb, indicating that V-Ti-Nb has dissolved into the M_7_C_3_ carbide with a length of 5.4 ± 2.1 μm and width of 1.3 ± 0.3 μm. The composition of Region L is similar to that of region C but is slightly rich in Si and poor in Ni, containing 0.1 at% V and 0.1 at% Zr. Its Vickers hardness is 464.8 HV, 7.3% higher than that of region L (430.7 HV), indicating that a small amount of V and Zr has solid-solution strengthened the A matrix with a size of 18.5 ± 13.3 μm. Strong carbide-forming elements (Ti, Zr, V, Nb) can shift the C curve to the right, improving the hardenability of the HCCI alloy. The V and Zr dissolved into austenite improve the hardenability of the HCCI-VFC alloy. In short, compared to Figure 1e, the dissolved alloying elements did not refine the size of M_7_C_3_ carbide and austenite in as-cast HCCI alloy but improved the hardness of the matrix.

Figure 1j–l shows the SEM images of HCCI-VFC after heat treatment. Region M is the M matrix with a composition of (Fe_0.82_Cr_0.13_M_0.04_(VFC)_0.01_)C_x_ (M = Si, Mn, Mo, Ni, (VFC)_0.01_ = V_0.007_Ti_0.003_). V-Ti-Nb and B elements are mainly dissolved in M_7_C_3_ carbides, while Zr is dissolved in austenite and martensite. The Vickers hardness of the martensite in Figure 1l is 763.2 HV, which is 7.0% higher than that of the matrix in region F. This further proves that the matrix hardness is improved by solution strengthening. Additionally, the eutectic carbide has transformed into clump- and worm-shape. Region N is M_7_C_3_ carbide with a composition of (Cr_0.50_Fe_0.42_M_0.05_(VFC)_0.03_)_7_C_3_ (M_0.04_ = Si_0.007_Mn_0.022_Mo_0.004_Ni_0.007_), (VFC)_0.063_ = V_0.047_Ti_0.010_Nb_0.006_). Figure 1l shows that white secondary carbides are dispersed on the HCCI-VFC matrix. After HCCI are tempered, carbon and chromium in residual austenite will precipitate in secondary carbides [23].

Thus, the alloying process of adding VFC ingot to HCCI alloy can be described as follows: first, we melt alloy elements into the VFC alloy (Figure 2a). When the HCCI alloy was completely melted in the smelting furnace, VFC ingot was added into the HCCI melt (Figure 2b). After that, V-Ti-Nb alloying elements are rapidly dispersed in the melt through electromagnetic stirring (Figure 2c). During pouring, V-Ti-Nb-Fe-Zr-C-B alloying elements were dissolved into the austenite and M_7_C_3_ carbide (Figure 2d).

### 3.2. Phase Transformation

Figure 3 and Table 2 show the XRD patterns and refinement results of the tempered HCCI and HCCI-VFC alloys. The XRD results are refined by the Rietveld method using the FullProf software. The phase content of HCCI and HCCI-VFC is calculated directly using FullProf software. They are both composed of martensite, residual austenite, and M_7_C_3_ carbide. Additionally, HCCI alloy comprises 70.7 wt% M, 6.0 wt% A’, and 23.3 wt% M_7_C_3_ carbide with a hexagonal structure. In HCCI-VFC alloy, the A’ content decreased to 0.9 wt%, and the M content increased to 82.5 wt%, indicating that VFC addition improves the hardenability of austenite, increases the martensite content, and reduces the A’ content. Notably, the HCCI-VFC alloy contains 7.9 wt% M_7_C_3_ with an orthogonal structure and 8.7 wt% M_7_C_3_ with a hexagonal structure. In other words, the VFC addition changed the crystallization process of the HCCI alloy. At the same time, the dissolved alloying elements expand the cell volume of M, A’, and M_7_C_3_ carbide.

Cr_7_C_3_ has two crystal structures, which are a hexagonal structure with a space group (SG) of P6_3_mc (186) at lower temperatures and an orthogonal structure with the Pnma (62) SG at high temperatures [24]. The formation enthalpy for Ortho-Cr_7_C_3_ and Hexa-Cr_7_C_3_ are −0.115 eV/atom and −0.087 eV/atom, respectively, so they are thermodynamically stable. Moreover, Ortho-Cr_7_C_3_ is more stable than Hexa-Cr_7_C_3_ [25]. Xing et al. [16] reported the chemical stability of (Cr, Fe)_7_C_3_ carbide at high temperatures could be improved by adding the alloying elements of Mo, W, and B into Cr-Ni cast irons to form (Cr, Fe, Mo, W)_7_(C, B)_3_ compound. Moreover, the added alloying elements of Mo and W partially replaced the Cr and Fe atoms to form stronger metal bonds, reduced the Gibbs free energy of the M_7_C_3_ carbide, improved the atomic binding energy in the carbide, and enhanced the chemical stability. In addition, the chemical bonds formed by B and (Cr, Mo, W) is more potent than those by C and (Cr, Mo, W), and the chemical stability of the formed M_7_(C, B)_3_ carbide is also greatly improved. Table 1 shows that V-Ti-Nb and B elements are mainly dissolved into M_7_C_3_ carbides to form the more stable (Cr, Fe, V, Ti, Nb)_7_(C, B)_3_ compound. In other words, they are first dissolved in Ortho-M_7_C_3_, which is stable at high temperatures, then part of Ortho-M_7_C_3_ is transformed into Hexa-M_7_C_3_, which is stable at low temperatures, thereby forming two M_7_C_3_ structures with similar content in HCCI-VFC alloy. Compared to HCCI alloy, it can be concluded that V-Ti-Nb-B addition can stabilize Ortho-M_7_C_3_ carbide. The solubility of Zr in α-Fe and γ-Fe is 0.1 at% and 0.2 at%, respectively.

Moreover, we only added 3.1 wt% VFC alloy, equivalent to only adding 0.07% Zr. Therefore, Zr can indeed be dissolved in γ-Fe and α-Fe and can also be dissolved in A and M. However, because Zr has the largest atomic radius in the alloy (0.145 nm), Zr is not conducive to stabilizing M_7_C_3_ compounds and tends to dissolve entirely in A and M, causing unit cell volume expansion. The atomic radii of V (0.135 nm) and Nb (0.148 nm) are also larger than those of Fe (0.127 nm) and Cr (0.127 nm), resulting in the volume expansion of the M_7_C_3_ cell in HCCI-VFC alloy.

### 3.3. Alloying Effect on Mechanical Properties

#### 3.3.1. Hardness

The hardness of the matrix and carbide in the alloy determines the hardness of high chromium cast iron. Table 3 shows HCCI-VFC alloy has a higher hardness than HCCI alloy. Moreover, the effect of alloying on improving the hardness of heat-treated alloys is more pronounced. EDS results in Table 1 show that V-Ti-Nb and B elements are mainly dissolved in M_7_C_3_ carbides, while a small amount of V and all Zr are dissolved in austenite and martensite, resulting in a solid solution strengthening effect. After quenching, most of A is transformed into M, increasing the hardness of HCCI and HCCI-VFC by 20.0% and 20.9%, respectively. After tempering, the A’ content decreases, and the granular secondary carbides precipitate on the matrix, increasing the hardness of HCCI and HCCI-VFC alloy by 4.5% and 6.4%, and matrix hardness by 3.3% and 3.6%, respectively. Hexa-Cr_7_C_3_ has a higher shear modulus (G = 117.2 GPa) and Young’s modulus (E = 311.2 GPa) than Ortho-Cr_7_C_3_ with G = 109.4 GPa and E = 293.6 GPa [25]. Therefore, changes in the M_7_C_3_ structure should have little impact on strength. The total M_7_C_3_ content in both structures is 16.6 wt%, less than 23.3 wt% in HCCI alloy. However, due to an increase of 11.5 wt% in the M content, a decrease of 5.1% in the A’ content, and an increase of more than 7% in the matrix hardness, the hardness of the HCCI-VFC alloy increased to 63.4 HRC, which is 4.1% higher than that of HCCI alloy.

#### 3.3.2. Toughness

The impact toughness of the heat-treated HCCI-VFC alloy increased from 7.1 J/cm^2^ of HCCI alloy to 8.3 J/cm^2^, increasing by 16.9%. High chromium cast iron is composed of M_7_C_3_ carbide and the M matrix and can be regarded as an in situ composite material in which M_7_C_3_ carbide is the reinforcing phase, and M is the matrix. Table 2 shows that the M content increased by 11.5 wt% and the M_7_C_3_ content decreased to 16.6 wt% in HCCI-VFC alloy, meaning adding VFC alloy weakens the continuity of carbide at grain boundaries and improves the impact toughness of HCCI-VFC.

To further analyze the reason for the improvement of impact toughness, Figure 4 shows the two-dimensional and three-dimensional impact fracture morphology of the heat-treated HCCI and HCCI-VFC alloys. The black region in Figure 4a,c represents the crack source, and the blue, green-yellow, and red-orange areas correspond to the fracture’s fiber area, radiation area, and shear lip area, respectively. The area ratios of the three areas were measured by Image-Pro-Plus software (version: November-2022), as shown in Table 4. The cracks in both alloys originate from the fiber zone. The fiber area ratio of HCCI-VFC alloy is 39.0% less than that of HCCI alloy, while the radiation area ratio is 217.9% higher than that of HCCI alloy. The propagation from the fiber area to the radiation area indicates a transition from slow crack growth to rapid unstable crack growth. The larger the radiation area ratio, the higher the toughness of the alloys. The final fracture area forms a shear lip. The shear lip area ratio of HCCI-VFC samples increased by 5.8%. When the material’s toughness is better and the crack propagation speed is lower, the shear lip area ratio in the fracture surface is more prominent. For example, the shear lip may disappear on the brittle material’s fracture. From the three fracture area ratios, the HCCI-VFC has more ductile fracture characteristics, improving impact toughness.

Figure 4b,d show the fracture height of the heat-treated HCCI-VFC alloy is higher than the HCCI alloy, meaning the HCCI-VFC alloy has a higher elongation along the longitudinal axis perpendicular to the fracture surface than that of HCCI.

Figure 4e,f shows the SEM images of the impact fracture surface corresponding to Figure 4a,d. The fracture morphology of HCCI alloy (Figure 4e) is mainly composed of river pattern and cleavage plane. The tearing fiber of martensite can be observed in region A, indicating that the martensite matrix is the main phase that bears impact force. Figure 4f shows tear edges in HCCI-VFC alloy, and the fracture morphology is significantly different from Figure 4e. For example, the carbides in region B are orderly and linearly distributed, surrounded by martensite. The level of martensite tearing is slightly lower than in Figure 4e. This result proves that the strengthened M_7_C_3_ carbide and the high content of the M matrix help the alloy absorb impact energy and improve the interfacial strength between the carbide and martensite, which is beneficial to improving the impact toughness of HCCI-VFC.

#### 3.3.3. Wear

The friction coefficient and wear loss of HCCI-VFC are 0.433 and 0.0185 g/h, smaller than the corresponding 0.443 and 0.0199g/h of HCCI and decreased by 2.3% and 7.0%, respectively. Figure 5a,b shows the SEM images of the wear surface. Many deep ditches, deep exfoliation (region A), deep scratches (region B), and rough scratches are found in HCCI alloy (Figure 5a). In Figure 5b, HCCI-VFC alloy has a fractured M_7_C_3_ carbide and deformed matrix (region C), small ditches, superficial peeling, and deep scratches (region D). The wear mechanism mainly behaves as abrasive wear and adhesive wear. The hardness of the carbide and matrix of HCCI-VFC is higher than that of HCCI, and HCCI-VFC has better toughness, which endows the HCCI-VFC alloy with better wear resistance and friction reduction.

Figure 5d,e shows the friction coefficient and wear loss curves versus time. The first stage is the run-in stage. As the contact area between the sample and the friction counterpart increases gradually, the friction coefficient increases gradually, and the wear loss of the sample also increases gradually. The second stage is the stable wear stage. The friction coefficient remains unchanged, and the wear loss increases almost linearly. The third stage is the severe wear stage. With an increase in wear time, the friction coefficient and the wear loss increase rapidly.

## 4. Conclusions

In this paper, the multi-element microalloying effect of adding 75.9V-12.6Fe-5Ti-3.1Nb-2.2Zr-0.6B-0.6C alloy ingot (VFC) into 2.99C-0.57Si-0.55Mn-17.76Cr-0.11Mo-0.12Ni (HCCI) was studied. The main conclusions are as follows:

(1) The added V-Ti-Nb-B are dissolved in M_7_C_3_ carbide to form the (Cr, Fe, V, Ti, Nb)_7_(C, B)_3_ alloy carbide, and a small amount of V and all Zr are dissolved in austenite and martensite. Adding VFC has little effect on M_7_C_3_ carbide and austenite or martensite size. Heat-treated HCCI and HCCI-VFC alloys are composed of martensite, residual austenite, and M_7_C_3_ carbide. After tempering, secondary carbides are dispersed and precipitated on the martensite matrix. Adding VFC into HCCI improved the hardenability of HCCI, decreased the residual austenite content from 6.0 wt% to 0.9 wt%, increased the martensite content from 70.7 wt% to 82.5 wt%, and changed the structure and content of M_7_C_3_ carbide from 23.3 wt% hexagonal M_7_C_3_ in HCCI to 7.9 wt% orthogonal M_7_C_3_ and 8.7 wt% orthogonal M_7_C_3_ carbide in HCCI-VFC.

(2) The solid-soluted V-Ti-Nb-B strengthened the M_7_C_3_ carbide, and the solid-soluted V-Zr strengthened austenite and martensite, increasing the hardness of as-cast and heat-tread HCCI alloy to 49.3 HRC and 63.4 HRC, increased the hardness of austenite and martensite to 464.8 HV and 763.2 HV, increased the impact toughness from 7.1 J/cm^2^ to 8.3 J/cm^2^, and reduced the friction coefficient and wear loss by 2.3% and 7.0%, respectively. The wear mechanism behaves as abrasive wear and adhesive wear. Thus, the hardness, toughness, wear resistance, and friction resistance of HCCI alloy have been improved simultaneously by adding a multi-element VFC alloy, showing a significant synergistic effect.

## Figures and Tables

**Figure 1 materials-16-03292-f001:**
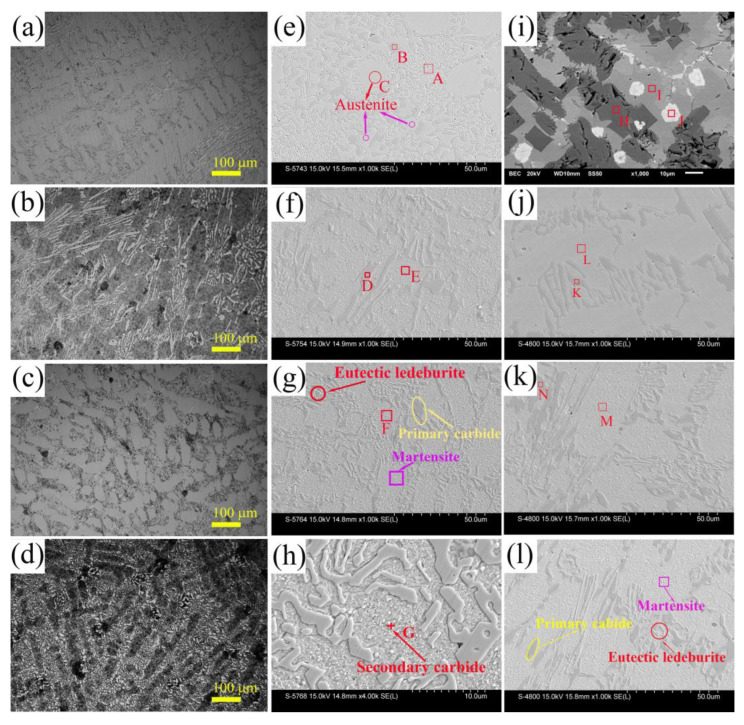
OM and SEM images of HCCI, VFC, and HCCI-VFC alloys. (**a**–**d**): OM images: (**a**) as-cast HCCI; (**b**) heat-treated HCCI; (**c**) as-cast HCCI-VFC; (**d**) heat-treated HCCI-VFC; (**e**–**l**): SEM images: (**e**) as-cast HCCI; (**f**) quenched HCCI; (**g**) tempered HCCI; (**h**) enlarged view of tempered HCCI; (**i**) VFC alloy; (**j**) as-cast HCCI-VFC; (**k**) quenched HCCI-VFC; (**l**) tempered HCCI-VFC.

**Figure 2 materials-16-03292-f002:**
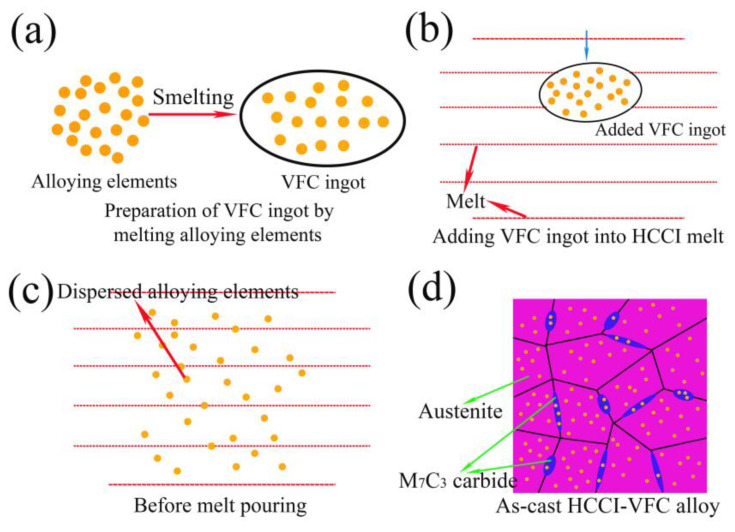
Melting micro-process models (**a**) melting alloy elements into VFC ingot; (**b**) adding VFC into HCCI melt; (**c**) VFC dispersing in the melt; (**d**) as-cast HCCI-VFC alloy.

**Figure 3 materials-16-03292-f003:**
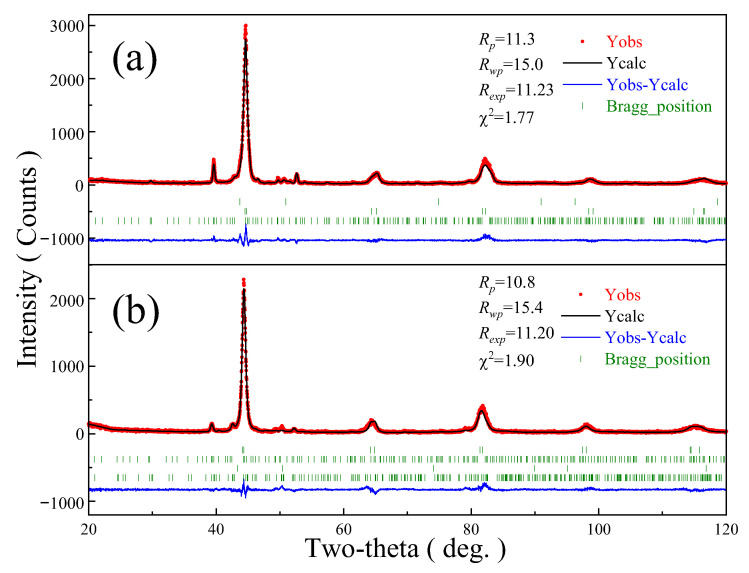
XRD patterns and refinement results of the heat-treated alloys. (**a**) HCCI; (**b**) HCCI-VFC.

**Figure 4 materials-16-03292-f004:**
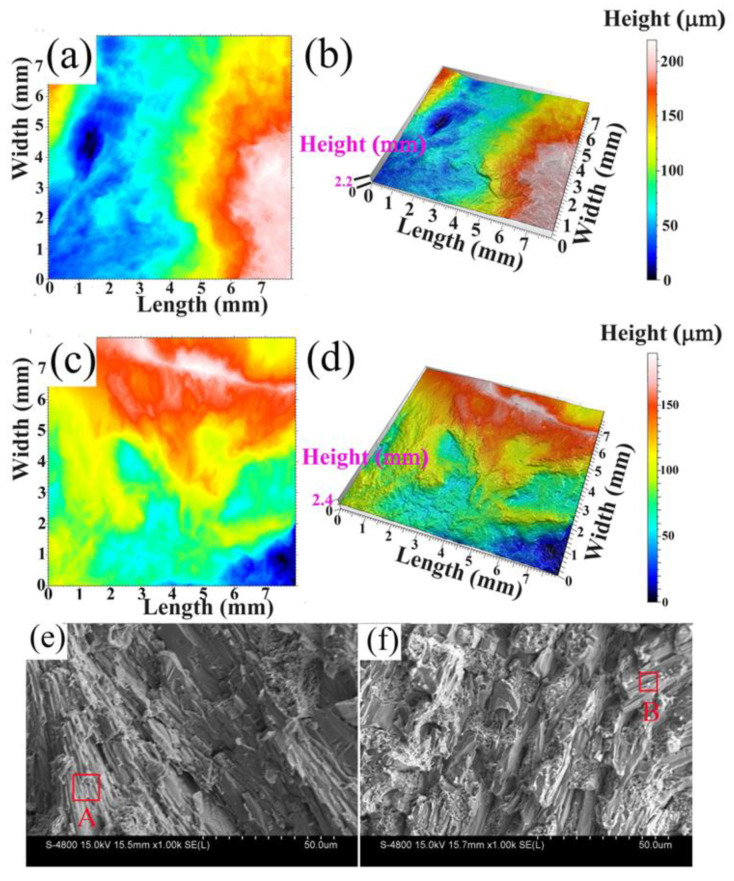
Macro morphology of impact fracture of the heat-treated HCCI and HCCI-VFC alloy. (**a**) two-dimensional fracture morphology of HCCI; (**b**) three-dimensional fracture morphology of HCCI; (**c**) two-dimensional fracture morphology of HCCI-VFC; (**d**) three-dimensional fracture morphology of HCCI-VFC; (**e**) microstructure of impact fracture of HCCI; (**f**) microstructure of impact fracture of HCCI-VFC.

**Figure 5 materials-16-03292-f005:**
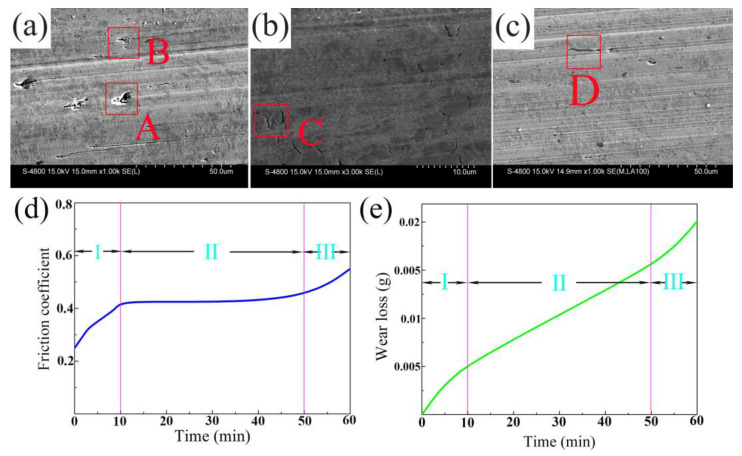
SEM images of the wear surface of heat-treated alloys and the friction coefficient and wear loss curves versus time. (**a**) wear surface of HCCI; (**b**) amplified view of (**a**); (**c**) wear surface of HCCI-VFC; (**d**) the friction coefficient curves versus time; (**e**) wear loss curves versus time.

**Table 1 materials-16-03292-t001:** EDS results of each region in Figure 1 (at%).

Element	Fe	Cr	Si	Mn	Mo	Ni	V	Ti	Nb	Zr	B	C	Possible Phases
A	24.9	27.6	0.2	1.2	0	0.2	0	0	0	0	0	45.9	M_7_C_3_
B	24	26	1.1	1.1	0.2	0.7	0	0	0	0	46.9	0	M_7_C_3_
C	58.1	9.1	0.8	0.8	0	1.1	0	0	0	0	0	30.1	A
D	24.2	28.6	0	1.4	0.3	0.8	0	0	0	0	0	44.7	M_7_C_3_
E	50.9	8.5	0.9	1.3	0	0.9	0	0	0	0	0	37.5	M
F	52.3	8.3	0.7	1.4	0	1	0	0	0	0	0	36.3	M
G	17.41	4.56	0.22	0.45	0.12	0.26	0.28	0.16	0	0	50.75	25.72	M_7_C_3_
H	0.7	0	0	0	0	0	68.8	3.9	1.5	0.1	0	25.0	(V, M)_2_C
I	36.2	0	0	0	0	0	54.0	0.5	0.9	0.1	8.3	0	V(Fe, M)
J	0.6	0	0	0	0	0	3.4	11.1	1.5	29.9	35.3	18.2	(Zr, M)(C, B)
K	9.4	11.9	0.3	0.5	0	0.2	0.5	0.1	0.1	0	44.9	32.1	M_7_C_3_
L	24.5	3.8	0.3	0.4	0	0.1	0.1	0	0	0.1	16.2	54.5	A
M	23.5	3.7	0.2	0.4	0	0.2	0.2	0.1	0	0.1	19.8	51.8	M
N	10.1	11.9	0.2	0.6	0.1	0.2	0.6	0.1	0.1	0	45.9	30.2	M_7_C_3_

**Table 2 materials-16-03292-t002:** Refinement results corresponding to Figure 2.

Phase/System/SG	Parameters	HCCI	HCCI-VFC
Martensite	*a* (nm)	0.286250(13)	0.289869(17)
/Tetragonal	*c* (nm)	0.289249(20)	0.287309(18)
/I4/mmm (139)	*V* (×10^−3^ nm^3^)	23.7	24.1
	Content (wt%)	70.7	82.5
Austenite	*a* (nm)	0.35829(7)	0.3617(2)
/Cubic	*V* (×10^−3^ nm^3^)	45.9	47.3
/Fm$\bar{3}$m (225)	Content (wt%)	6.0	0.9
M_7_C_3_	*a* (nm)	—	0.45332(9)
/Orthogonal	*b* (nm)	—	0.7014(3)
/Pnma (62)	*c* (nm)	—	1.189(2)
	*V* (×10^−3^ nm^3^)	—	378.2
	Content (wt%)	—	7.9
M_7_C_3_	*a* (nm)	1.39084(7)	1.4008(2)
/Hexagonal	*c* (nm)	0.45028(7)	0.454385(0)
/P6_3_mc (186)	*V* (×10^−3^ nm^3^)	754.3	772.1
	Content (wt%)	23.3	8.7

**Table 3 materials-16-03292-t003:** Rockwell hardness (HRC) of high chromium cast iron alloy and Vickers hardness (HV) of the matrix in HCCI and HCCI-VFC alloys.

Sample	Alloy Hardness (HRC)			Matrix Hardness (HV)		
	As-Cast	As-Quenched	Tempered	As-Cast	As-Quenched	Tempered
HCCI	48.6 ± 0.2	58.3 ± 0.4	60.9 ± 0.3	430.7 ± 4.1	690.3 ± 7.3	713.4 ± 5.4
HCCI-VFC	49.3 ± 0.6	59.6 ± 0.3	63.4 ± 0.2	464.8 ± 4.7	736.7 ± 7.4	763.2 ± 5.6
Increased by (%)	1.4	2.2	4.1	7.9	6.7	7.0

**Table 4 materials-16-03292-t004:** Impact toughness (J/cm^2^) and composition of impact fracture of heat-treated alloys.

Sample	Impact Toughness (J/cm^2^)	Composition of Impact Fracture			
		Fiber Zone Ratio (%)	Radiation Zone Ratio (%)	Shear Lip Zone Ratio (%)	Fracture Height (mm)
HCCI	7.1	55.1	8.9	36.0	2.2
HCCI-VFC	8.3	33.6	28.3	38.1	2.4
Increased by (%)	16.9	-39.0	218.0	5.8	9.1

## Data Availability

All data included in this study are available upon request by contact with the corresponding author.

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
