# Peer review of "Effect of Multi-Element Microalloying on the Structure and Properties of High Chromium Cast Iron"

_materials, 2023, doi:10.3390/ma16093292_

Round 1

Reviewer 1 Report

The article is written at a fairly high level: a detailed review of the literature was made, and the methodology was described.

The authors comprehensively and diversified studied the properties and structure of the material; however, hardness, impact strength, and tribotechnical properties were taken as the studied properties. Why weren't fractograms of sample surfaces studied after impact tests?

It would also be good to increase the article value by studying and presenting the result of the influence of additives on the tensile strength of alloys.

Author Response

Replies to the Review Reports about the Manuscript of

Materials-2352825

Dear Ms. Irina Mariana Sandulache, Editors and reviewers,

Firstly, we take the opportunity to express our gratitude to the editors and reviewers for taking so much time to evaluate our work. In particular, the reviewers have proposed professional suggestions, which make our work more complete. Besides answering the questions proposed by the editors and reviewers, we have overhauled this article and sincerely hope to get a positive reply from you.

Reviewers' comments:

Reviewer #1:

The article is written at a fairly high level: a detailed review of the literature was made, and the methodology was described. The authors comprehensively and diversified studied the properties and structure of the material.  

Answer to Reviewer #1: Thank you for your valuable time reviewing and recognizing our work!

Q1: however, hardness, impact strength, and tribotechnical properties were taken as the studied properties. Why weren't fractograms of sample surfaces studied after impact tests?

Answer to Q1: We agree with your suggestion. We believe hardness, impact strength, and tribological properties are the primary performance affecting high chromium cast iron use. Therefore, this article provides a detailed analysis of the fracture morphology of HCCI and HCCI-VFC impact samples in Figure 4. Figure 4a-d shows the macroscopic fracture morphology captured by the laser confocal microscope. From the three fracture area ratios, the HCCI-VFC has more ductile fracture characteristics, improving impact toughness. Figure 4e-f shows the microscopic fracture morphology captured by the scanning electron microscope. They also show that HCCI-VFC has more ductile fracture characteristics. For example, the tear level of martensite is lower than that of HCCI.

Modification: The relevant content has been adjusted.

Q2: It would also be good to increase the article value by studying and presenting the result of the influence of additives on the tensile strength of alloys.

Answer to Q2: You are very professional. As you know, high chromium cast iron has high hardness and excellent wear resistance, which are closely related to the tensile strength of the alloy. Therefore, we also plan to study the effect of the added alloying elements on tensile strength. However, we found that the hardness of the alloy was too high to fix the sample with a fixture during the test. The sample will gradually detach from the fixture so that the sample cannot be pulled apart. Therefore, we have to use hardness to characterize wear resistance.

Modification: This section has not been modified.

This concludes the reply of the reviewer.

All modifications have been highlighted in the revised manuscript.

Thank you again for your hard work!

Sincerely yours,

Tao Liu, Ji-Bing Sun, Zhi-Xia Xiao, Jun He, Wei-Dong Shi, and Chun-Xiang Cui

Reviewer 2 Report

The presented manuscript includes the study of the effect of multi-element microalloying on the structure and properties of high chromium cast iron. Paper fit to the Materials journal.

The results of the work are presented on a good level. Paper well written and structured.

Just small engineering correction can be mentioned: “30 mm x 7 mm x 6 mm” – in engineering drawings all sizes 

Author Response

Replies to the Review Reports about the Manuscript of

Materials-2352825

Dear Ms. Irina Mariana Sandulache, Editors and reviewers,

Firstly, we take the opportunity to express our gratitude to the editors and reviewers for taking so much time to evaluate our work. In particular, the reviewers have proposed professional suggestions, which make our work more complete. Besides answering the questions proposed by the editors and reviewers, we have overhauled this article and sincerely hope to get a positive reply from you.

Reviewers' comments:

Reviewer #2:

The presented manuscript includes the study of the effect of multi-element microalloying on the structure and properties of high chromium cast iron. Paper fit to the Materials journal. The results of the work are presented on a good level. Paper well written and structured. Just small engineering correction can be mentioned: "30 mm x 7 mm x 6 mm" – in engineering drawings all sizes.

Answer to Reviewer #2: We agree with your suggestion. According to GB/T3960-2016 standard, the wear samples were processed to 30 mm × 7 mm × 6 mm, and the M-200 model friction and wear testing machine was used to conduct the wear resistance test. Of course, as you mentioned, "30 mm × 7 mm × 6 mm" can be applied to engineering drawings of all sizes.

Modification: According to your suggestion, the relevant content has been modified to "30 mm × 7 mm × 6 mm" – in engineering drawings of all sizes.

This concludes the reply of the reviewer.

All modifications have been highlighted in the revised manuscript.

Thank you again for your hard work!

Sincerely yours,

Tao Liu, Ji-Bing Sun, Zhi-Xia Xiao, Jun He, Wei-Dong Shi, and Chun-Xiang Cui

Reviewer 3 Report

I personally feel there is no significant novelty in the present paper as there are many similar papers are published already. Only difference in the present paper is few micro alloying elements are different and the quality of the microstructure is not good and does not depicts the much information. Authors also not discussed about the phase diagram for all the added micro alloying elements. It is quite a difficult to discuss the phase diagram for multi elemental system. The XRD of both samples looks similar except the crystallinity, both the compositions show similar phases, no change in the phases. Only clarity I got is the differences in the hardness after adding the multi elements. That is acceptable and expected. But, Authors failed completely in discussing the microstructure in detail. Even though article sounds good, but not suitable for the publication in reputed journals like Materials. Still more information regarding the phase diagrams, optical microstructures, phase analysis is required. Quality of the figures are poor.

Author Response

Replies to the Review Reports about the Manuscript of

Materials-2352825

Dear Ms. Irina Mariana Sandulache, Editors and reviewers,

Firstly, we take the opportunity to express our gratitude to the editors and reviewers for taking so much time to evaluate our work. In particular, the reviewers have proposed professional suggestions, which make our work more complete. Besides answering the questions proposed by the editors and reviewers, we have overhauled this article and sincerely hope to get a positive reply from you.

Reviewers' comments:

Reviewer #3:

I personally feel there is no significant novelty in the present paper as there are many similar papers are published already. Only difference in the present paper is few micro alloying elements are different and the quality of the microstructure is not good and does not depict the much information. Authors also not discussed about the phase diagram for all the added micro alloying elements. It is quite a difficult to discuss the phase diagram for multi elemental system. The XRD of both samples looks similar except the crystallinity, both the compositions show similar phases, no change in the phases. Only clarity I got is the differences in the hardness after adding the multi elements. That is acceptable and expected. But authors failed completely in discussing the microstructure in detail. Even though article sounds good, but not suitable for the publication in reputed journals like Materials. Still more information regarding the phase diagrams, optical microstructures, phase analysis is required. Quality of the figures are poor.

Answer to Reviewer #3: We admire you for taking time out of your busy schedule to read our paper carefully. Thank you for your suggestions.

(1) About novelty: Perhaps the innovation of this article is not as surprising, but we believe that the innovation of our work mainly focuses on the following aspects: (a) Few people have discovered and discussed the role of Zr and the impact of alloying elements on the crystal structure of M7C3. (b) Although the study of multi-element addition is complex, we tried to carry out such work. So, multi-element 75.9V-12.6Fe-5Ti-3.1Nb-2.2Zr-0.6C-0.6B alloy is added to HCCI to explore multiple synergistic strengthening effects. We found that the added V-Ti-Nb-B are dissolved in M7C3 carbide to form the (Cr, Fe, V, Ti, Nb)7(C, B)3 alloy carbide, and a small amount of V and all Zr are dissolved in austenite and martensite. Adding alloying elements to HCCI increased hardness by 4.1 %, increased toughness by 16.9 %, and decreased wear loss by 7.0 %.

(2) About the phase diagram: As you mentioned, it is pretty challenging to discuss the phase diagram for the multi-elemental system. But it is because of its difficulties that we need to try to discover solutions from an experimental perspective, which is also the primary purpose of this article.

(3) About XRD patterns: You said that the XRD patterns of both samples look similar except for the crystallinity. Both compositions show similar phases, with no change in the phases. The fact is indeed like this. However, the small changes in XRD images are often difficult to distinguish with the eyes, so they must be identified through Rietveld calculations. In fact, the phase composition of the two samples is different, including the phase's relative content and crystal structure.

(4) About more information regarding the phase diagrams, optical microstructures, and phase analysis: the phase diagram of high chromium cast iron has been calculated by Ref.[9]. Now, we added the optical microstructure of VFC alloy (Figure.1 i) and high chromium cast iron (Figure.1 a-d) and the corresponding phase analysis.

(5) About the quality of figures: Perhaps due to a decrease in the resolution after combining, we have now increased the pixel size of the images.

Modification: Some analysis and images have been added.

This concludes the reply of the reviewer.

All modifications have been highlighted in the revised manuscript.

Thank you again for your hard work!

Sincerely yours,

Tao Liu, Ji-Bing Sun, Zhi-Xia Xiao, Jun He, Wei-Dong Shi, and Chun-Xiang Cui

Reviewer 4 Report

The manuscript discusses the effects of addition of a V-Fe-Ti-Nb-Zr-B-C ingot on the microstructure and mechanical properties of a high chromium cast iron. The results are interesting and can find industrial applications. However, the following comments need to addressed before a decision on whether to accept the manuscript is made.

 1- The English is clear but can be improved.

2- Please discuss the reasons for the stoichiometry of the selected VFC alloy, i.e. 75.9V-12.6Fe-5Ti-3.1Nb-2.2Zr-0.6C-0.6B?

3- Please discuss the reason for the selected VFC%, i.e. 3.1 wt%? and why other percentages were not tested?

4- Is there any references for the applied heat treatment cycles of the alloys?

5- What was the counterpart in the wear tests?

6- Please show the microstructure of the VFC alloy before addition to the melt and identify different phases in the microstructure?

7- Fig. 1-a needs to be replaced. The microstructural features in the figure is not clear at all. Also, Figs. a-c and e-g may be enlarged.

8- I think the labels and arrows in Fig1-i need to be repositioned to show the concept more clearly.   

9- Considering the scale bars on the SEM micrographs, the microstructural features can be shown by optical microscopy as well. I strongly recommend inclusion of the OM micrographs.

10- Page 4- Line 141: What are the white granular secondary carbides shown in Figure 1c-d?

11- Page 4- Line 159: Please explain why B elements are mainly dissolved in M7C3 carbides, while Zr is dissolved in A and M phases?

12- Can you discuss the effects of VFC addition on continuity of the carbide phases at grain boundaries?

13- Page 5:  Please explain in section 2 how the percentages of different phases were calculated from the XRD spectra?

14- Page 5-Lines 176-181: What is the scientific explanation for improvement in the hardenability by VFC addition?

15- Table 3: The increase in the as-quenched alloy hardness by VFC addition is 2.2% not 7.9%!

16- The differences in alloy hardness in the as-cast and as-quenched conditions is very small and can be within the experimental measurement error of the equipment!

17- Fig. 3: Please encircle the fiber, radiation and shear slip zones in Figs. 3-e and 3-f.

18- Page 8-line 277-278: Please clarify that the samples have been heat treated.

19- Page 8: The friction coefficient and wear loss curves versus distance for each sample needs to be shown.

20- Page 8- line 283: For adhesive wear, the surface of the counterpart needs to be shown and the adhered material be analyzed.

The English is clear but can be improved.

Author Response

Replies to the Review Reports about the Manuscript of

Materials-2352825

Dear Ms. Irina Mariana Sandulache, Editors and reviewers,

Firstly, we take the opportunity to express our gratitude to the editors and reviewers for taking so much time to evaluate our work. In particular, the reviewers have proposed professional suggestions, which make our work more complete. Besides answering the questions proposed by the editors and reviewers, we have overhauled this article and sincerely hope to get a positive reply from you.

Reviewers' comments:

Reviewer #4:

The manuscript discusses the effects of addition of a V-Fe-Ti-Nb-Zr-B-C ingot on the microstructure and mechanical properties of a high chromium cast iron. The results are interesting and can find industrial applications. However, the following comments need to address before a decision on whether to accept the manuscript is made.

Answer to Reviewer #4: Thank you very much for affirming our work.

Q1: The English is clear but can be improved.

Answer to Q1: You revised it very carefully. Some professional words may be improper. We asked people who are good at English to revise it again, which seems to be improved.

Modification: Some sentences and vocabulary have been adjusted.

Q2: Please discuss the reasons for the stoichiometry of the selected VFC alloy, i.e. 75.9V-12.6Fe-5Ti-3.1Nb-2.2Zr-0.6C-0.6B?

Answer to Q2: Following your suggestion, we have further explained the element composition of the VFC alloy. The proportion of elements in VFC alloy is determined by the proportion of added elements in high chromium cast iron of relevant references. V is the most effective alloying element. According to Ref. [5], we added a high content of V (2.29 wt. %) and a small amount of C to the VFC alloy, hoping to compare the effect of MC and M2C phases formed by adding V alone in the ingot. Kopycinski et al. [9] added 0-0.2 wt.% Ti to high chromium cast iron (Cr 21.2 wt.%, C 1.7 wt.%). The bending strength of as-cast high chromium cast iron increased from 821MPa to 995MPa with the addition of 0.1 wt. % Ti, and decreased to 987 MPa with the addition of 0.2 wt.% Ti. Maja et al. [10] added 0.1-0.4 wt.% Nb to high chromium cast iron (Cr 10-30 wt. %, C 2-3.5 wt. %) and observed that the hardness of cast high chromium cast iron increased from 432HB to 485HB. However, adding more Nb (0.4-0.6 wt.%) decreased the hardness of as-cast high chromium cast iron to 438HB. Therefore, we add a small amount of Ti (0.15 wt.%) and Nb (0.10 wt.%) to improve the alloying effect of high chromium cast iron by solid solution in VC or V2C. The VFC alloy content added in high chromium cast iron is 3.1 wt.%, equivalent to adding 2.29 wt.% V, 0.15 wt.% Ti, and 0.10 wt.% Nb. The proportion is roughly equivalent to the alloy element content of 2.15 wt.% added in Ref. [9]. Zr (0.07wt.%) has a stronger ability to form carbide than Ti, Nb, V, Cr, and Fe and can also deoxygenate and purify molten steel. B (0.02wt.%) [16] can solidly dissolve into M7C3 carbides to form M7(C, B)3, which improves the hardness of carbide in high chromium cast iron. According to the content added in Refs. [12] and [16], Zr and B only need shallow content to bring a noticeable effect, so the content added is negligible.

Modification: We added relevant literature and the corresponding statement.

Added references:

[9] Kopyciński, D.; Siekaniec, D.; Szczęsny, A.; Guzik, E.; Nowak, A. The effect of Fe-Ti inoculation on solidification, structure and mechanical properties of high chromium cast iron[J]. Arch Metall Mater, 2017, 62, 2183-2187. http:doi.org/10.1515/amm-2017-0321.

[10] Maja, M.E.; Maruma, M.G.; Mampuru, L.A. Effect of niobium on the solidification structure and properties of hypoeutectic high-chromium white cast irons[J]. J S Afr I Min Metall, 2016, 116, 981-986. http:doi.org/10.17159/2411-9717/2016/v116n10a14.

Q3: Please discuss the reason for the selected VFC%, i.e. 3.1 wt.%? and why other percentages were not tested?

Answer to Q3: The weight ratio of the VFC alloy to high chromium cast iron is obtained by adding the required weight ratio of various elements for alloying high chromium cast iron. 3.1 wt % VFC = 2.29V-0.46Fe-0.15Ti-0.10Nb-0.07Zr-0.02C-0.02B (wt. %). The previous question has explained the proportion of each element in the VFC alloy.

This article attempts to explore the synergistic strengthening effect of various alloying elements on high chromium cast iron by exploring the improvement of the microstructure and mechanical properties of high chromium cast iron by adding multiple trace alloying elements. So this paper only discusses the alloy ingot with one component.

Modification: This section has not been modified.

Q4: Is there any references for the applied heat treatment cycles of the alloys?

Answer to Q4: Thank you for your professional advice! We have indeed determined the heat treatment process based on the literature. Gong et al. [19] believe that the best heat treatment temperature for high chromium cast iron depends on the specific composition. The quenching temperature is about 1000 ℃, and the best holding time is 1-3 h. Appropriate tempering temperature and holding time can increase the toughness and wear resistance of high chromium cast iron. Liu et al. [20] studied the effect of quenching temperature on the size of M7C3 carbides in high chromium cast iron (17.4 wt% Cr, 4wt% C). lower quenching temperatures and shorter holding time, such as at 900 °C for 2 h, increase the number of fine secondary M7C3 carbides; However, the number of coarse M7C3 carbides larger than 11.2 μm increases in case of higher quenching temperatures and longer holding times, such as at 1050 °C for 6 h. Chen et al. [21] investigated the effect of heat treatment temperature on the mechanical properties of hyper eutectic high chromium cast iron (19 wt% Cr, 3.9 wt% C). The hardness and wear resistance are the best when quenched at 1000 °C. When the tempering temperature is 500 °C, the hyper eutectic HCCI shows the best wear resistance. When tempered at 400 °C, hyper eutectic HCCI shows the highest hardness. Zhi et al. [22] studied the effect of quenching temperature (850-1050 ℃) on hyper eutectic high chromium cast iron (18wt % Cr, 4wt % C). With increasing the quenching temperature, the bulk hardness and matrix microhardness increased and reached peak values of 64.6 HRC and 850 HV, respectively, at 1000 °C. But they all decreased at 1050 °C.

In summary, according to the different Cr content (12-28 wt.%), the best quenching temperature for high chromium cast iron is between 950 ℃ and 1050 ℃. To eliminate the residual stress during quenching, high-chromium cast iron should be tempered as soon as possible. Compared with low-temperature tempering (250 ℃), secondary carbide in high chromium cast iron tempered at medium to high temperatures (400 ℃ -500 ℃) is more likely to precipitate from residual austenite, and the hardness of the alloy is improved through dispersion strengthening. Based on the above literature, alloys containing 2.76% C and 17.91% Cr in this article should use higher quenching temperatures and medium temperature tempering, i.e., the quenching process should be 960℃×3h, and the tempering process should be 450℃×2h.

Modification: Since heat treatment is not the main topic of this paper, this expression is added to the revised manuscript: according to the heat treatment processes in references [19-22], for high chromium cast iron containing 2.76% C and 17.91% Cr, the commonly used quenching process is 960 ℃ × 3 h, tempering process is 450 ℃ × 2h.

Added references:

[19] Gong, L.; Fu, H.; Zhi, X. Corrosion wear of hypereutectic high chromium cast iron: a review[J]. Metals, 2023, 13, 308-319. http:doi.org/10.3390/met13020308.

[20] Liu, Q.; Zhang, H.; Wang, Q.;Nakajima, K. Effect of heat treatment on microstructure and mechanical properties of Ti-alloyed hypereutectic high chromium cast iron[J]. ISIJ Int, 2012, 52, 2288-2294. http:doi.org/10.2355/isijinternational.52.2288.

[21] Chen, Z.; Guo, Q.; Fu, H.; Zhi, X. Effect of heat treatment on microstructure and properties of modified hypereutectic high chromium cast iron[J]. Mater Test, 2022, 64, 33-54. http:doi.org/doi:10.1515/mt-2021-2010.

[22] Zhi, X.H.; Xing, J.D.; Gao, Y.M.; Fu, H.G.; Peng, J.Y.; Xiao, B. Effect of heat treatment on microstructure and mechanical properties of a Ti-bearing hypereutectic high chromium white cast iron[J]. Mater. Sci. Eng. A, 2007, 487, 171-179. http:doi.org/10.1016/j.msea.2007.10.009.

Q5: What was the counterpart in the wear tests?

Answer to Q5: We use GCr15 bearing steel as the counterpart's material in the sliding friction and wear test.

Modification: We added the counterpart materials in the Materials and Methods section.

Q6: Please show the microstructure of the VFC alloy before addition to the melt and identify different phases in the microstructure?

Answer to Q6:According to your suggestion, we have added the SEM image of the VFC alloy and analyzed the microstructure of VFC.

Modification: The added explanation has been added to section 3.1. Figure 1i shows the SEM image of the VFC alloy. The ingot shows three contrast. Black region H is riched in V, Ti, and C, and its composition is (V68.8Zr0.1Ti3.9Fe0.7Nb1.5)C25.0=(V, M)3C. V-C compounds have no stable V3C structure, while V2C, Fe2C, and Nb2C have similar stable M2C structures. Therefore, the added trace elements can dissolve in the V2C structure to form a (V, M)2C alloy solid solution. Gray region I is rich in V and Fe, corresponding to the V (Fe, M) matrix. White region J is (V3.4Zr29.9Ti11.1Fe0.6Nb1.5)(B18.2C35.3)=(Zr, M)0.9(B, C), riched in Zr, B, and C. ZrC, VC, TiC, and NbC can form MC structures. Therefore, V, Ti, and Nb replace Zr, and B replaces C to form the (Zr, M)(B, C) phase.

Q7: Fig. 1-a needs to be replaced. The microstructural feature in the figure is not clear at all. Also, Figs. a-c and e-g may be enlarged.

Answer to Q7: The unclear Figure 1a you see is mainly due to its small size. We use the image for comparison, using the same magnification and scale for all the other images. Now we change the images' arrangement to 4 rows and 3 columns to enlarge the images.

Modification: The relevant images have been adjusted.

Q8: I think the labels and arrows in Fig1-i need to be repositioned to show the concept more clearly.

Answer to Q8: We add a new image (Figure 2a) based on the original Figure 1i and change the position of the arrows to more clearly express the process.

Modification: The relevant images have been adjusted.

Q9: Considering the scale bars on the SEM micrographs, the microstructural features can be shown by optical microscopy as well. I strongly recommend inclusion of the OM micrographs.

Answer to Q9: Following your suggestion, we added OM micrographs of high-chromium cast iron in Figure 1a-d. Compared with SEM, the microstructure of high chromium cast iron can now be more clearly displayed.

Modification: The images have been added to the new Figure 1a-d.

Q10: Page 4- Line 141: What are the white granular secondary carbides shown in Figure 1c-d?

Answer to Q10: Theoretically, it can be M3C-, M7C3-, or M23C6-type carbide. We have added EDS results to Table 1 to analyze the elemental content of secondary carbide. According to EDS results, the white granular secondary carbide should be M7C3 carbide.

Modification: The relevant modification has been added to section 3.1.

Q11:Page 4- Line 159: Please explain why B elements are mainly dissolved in M7C3 carbides, while Zr is dissolved in A and M phases?

Answer to Q11: According to the periodic table, B and C are adjacent and have similar atomic radii and physical and chemical properties. Ref. [14] show that the B atom can replace the C sites in M7C3 carbide to form M7(C, B)3 carbide.

The Fe-Zr phase diagram in Ref. [11] shows that the solubility of Zr in γ-Fe is 0.2 at%. Moreover, we only added 3.1 wt% VFC alloy, equivalent to only adding 0.07% Zr. Therefore, Zr can indeed be dissolved in γ-Fe and α-Fe and can also be dissolved in A and M.

Modification: Relevant modifications have been made in section 3.2

Q12:Can you discuss the effects of VFC addition on continuity of the carbide phases at grain boundaries?

Answer to Q12: Our experimental results indicate that adding VFC has a solid solution-strengthening effect on the matrix and M7C3 of high chromium cast iron. Moreover, according to the calculation results in Table 2, adding VFC increased the M content and decreased the M7C3 content, meaning adding VFC alloy weakens the continuity of carbide at grain boundaries, thereby improving the toughness of high chromium cast iron alloys.

Modification: The expression has been added to the revised manuscript:adding VFC alloy weakens the continuity of carbide at grain boundaries and improves the impact toughness of HCCI-VFC.

Q13:Page 5: Please explain in section 2 how the percentages of different phases were calculated from the XRD spectra?

Answer to Q13: The XRD results are refined by the Rietveld method using the FullProf software in section 2. The phase content of HCCI and HCCI-VFC can be calculated directly using FullProf software.

Modification: The above content has been added to section 3.2.

Q14:Page 5-Lines 176-181: What is the scientific explanation for improvement in the hardenability by VFC addition?

Answer to Q14: The material's continuous cooling transformation curve (C curve) is the basis for judging the hardenability strength. Strong carbide-forming elements (Ti, Zr, V, Nb) can shift the C curve to the right, improving the hardenability of the material. We found that a small amount of V and all Zr were dissolved in austenite and martensite. Therefore, it is mainly the V and Zr dissolved into austenite that improves the hardenability of the alloy.

Modification: The expression has been added to section 3.1: Strong carbide-forming elements (Ti, Zr, V, Nb) can shift the C curve to the right, improving the hardenability of the HCCI alloy. The V and Zr dissolved into austenite improve the hardenability of the HCCI-VFC alloy.

Q15: Table 3: The increase in the as-quenched alloy hardness by VFC addition is 2.2% not 7.9%!

Answer to Q15: We admire your professionalism and attention. Per your suggestion, we have revised 7.9% in Table 3 to 2.2%.

Modification: We have revised 7.9% in Table 3 to 2.2%.

Q16:The differences in alloy hardness in the as-cast and as-quenched conditions is very small and can be within the experimental measurement error of the equipment!

Answer to Q16: We highly appreciate your attitude towards data rigor. Before testing hardness, we used a standard hardness block (45 HRC) to correct the HR-150A instrument to ensure its test error was less than 0.1 HRC. At the same time, we randomly took five points for each sample to average its hardness and calculate the standard deviation through the measured value to avoid an operational error. Therefore, although the hardness difference between some samples is minimal, we can guarantee that these data are trustworthy.

But you mentioned that the differences in alloy hardness in the as-cast and as-queued conditions are very small; it doesn't seem like that. The hardness difference between the as-cast and quenched samples is quite significant. The difference between the quenched and tempered samples is relatively small, but these differences are much more significant than the standard deviation.

Modification: This section has not been modified.

Q17:Fig. 3: Please encircle the fiber, radiation and shear slip zones in Figs. 3-e and 3-f.

Answer to Q17: Your questions are very constructive. The laser confocal microscope can distinguish the three areas of the fracture and display different colors by distinguishing the height of each area. Therefore, artificially highlighting these areas is no longer necessary, unlike the images captured by metallographic and scanning electron microscopes that require manual recognition.

To analyze possible fracture mechanisms, the SEM image of the fracture was magnified 1000 times (Figures 4e-f). Therefore, it is difficult for us to distinguish fiber, radiation, and shear lip areas.

Modification: This section has not been modified.

Q18:Page 8-line 277-278: Please clarify that the samples have been heat treated.

Answer to Q18: You are very professional and patient, and we have supplemented relevant explanations in section 3.3.

Modification: We have supplemented the relevant content in section 3.3.

Q19:Page 8: The friction coefficient and wear loss curves versus distance for each sample needs to be shown.

Answer to Q19: You have provided many valuable suggestions for our paper. According to your suggestion, we have added the friction coefficient and wear loss curves versus time in Figure 4d-e.

Modification: We have added the friction coefficient and wear loss curves versus time in Figure 4d-e and the corresponding explanation.

Q20:Page 8- line 283: For adhesive wear, the surface of the counterpart needs to be shown and the adhered material be analyzed.

Answer to Q20: The surface of the friction counterpart should be analyzed, as you mentioned. We were not aware of this problem at that time, and these friction counterparts were polished after the experiment to be used in the following experiment. We mainly judge the wear mechanism based on the surface of the sample. It seems that the evidence is not so sufficient.

We are sorry we can't add it anymore.

Modification: This section has not been modified.

This concludes the reply of the reviewer.

All modifications have been highlighted in the revised manuscript.

Thank you again for your hard work!

Sincerely yours,

Tao Liu, Ji-Bing Sun, Zhi-Xia Xiao, Jun He, Wei-Dong Shi, and Chun-Xiang Cui

Round 2

Reviewer 3 Report

Thank you for responding to my queries. I am now fully satisfied with your comments and the revisions.

Reviewer 4 Report

It seems that the authors have done a good job of revising the manuscript. I am not fully satisfied with some of the answers but believe that the manuscript can be accepted in its present form. 

Satisfactory